# Atlas of PD-L1 for Pathologists: Indications, Scores, Diagnostic Platforms and Reporting Systems

**DOI:** 10.3390/jpm12071073

**Published:** 2022-06-29

**Authors:** Stefano Marletta, Nicola Fusco, Enrico Munari, Claudio Luchini, Alessia Cimadamore, Matteo Brunelli, Giulia Querzoli, Maurizio Martini, Elena Vigliar, Romano Colombari, Ilaria Girolami, Fabio Pagni, Albino Eccher

**Affiliations:** 1Department of Diagnostic and Public Health, Section of Pathology, University of Verona, 37100 Verona, Italy; stefano.marletta@univr.it (S.M.); claudio.luchini@univr.it (C.L.); matteo.brunelli@univr.it (M.B.); 2Department of Pathology, Pederzoli Hospital, 37019 Peschiera del Garda, Italy; 3Division of Pathology, IEO, European Institute of Oncology IRCCS, Department of Oncology and Hemato-Oncology, University of Milan, 20139 Milan, Italy; nicola.fusco@unimi.it; 4Department of Molecular and Translational Medicine, University of Brescia, 25121 Brescia, Italy; enrico.munari@unibs.it; 5Section of Pathological Anatomy, School of Medicine, United Hospitals, Marche Polytechnic University, 60131 Ancona, Italy; a.cimadamore@staff.univpm.it; 6Department of Pathology and Diagnostics, University and Hospital Trust of Verona, 37126 Verona, Italy; giulia.querzoli@aovr.veneto.it; 7Department of Human Pathology of the Adult and Developmental Age “Gaetano Barresi”, University of Messina, 98124 Messina, Italy; mmartini@unime.it; 8Department of Public Health, University of Naples Federico II, 80100 Naples, Italy; elena.vigliar@unina.it; 9Unit of Surgical Pathology, Ospedale Fracastoro, 37047 San Bonifacio, Italy; romano.colombari@aulss9.veneto.it; 10Division of Pathology, Bolzano Central Hospital, 39100 Bolzano, Italy; ilaria.girolami@sabes.it; 11Department of Medicine and Surgery, Pathology, San Gerardo Hospital, University of Milano-Bicocca, 20900 Monza, Italy; fabio.pagni@unimib.it

**Keywords:** PD-L1, immunohistochemistry, immunotherapy, clones, scores

## Abstract

Background. Innovative drugs targeting the PD1/PD-L1 axis have opened promising scenarios in modern cancer therapy. Plenty of assays and scoring systems have been developed for the evaluation of PD-L1 immunohistochemical expression, so far considered the most reliable therapeutic predictive marker. Methods. By gathering the opinion of acknowledged experts in dedicated fields of pathology, we sought to update the currently available evidence on PD-L1 assessment in various types of tumors. Results. Robust data were progressively collected for several anatomic districts and leading international agencies to approve specific protocols: among these, TPS with 22C3, SP142 and SP263 clones in lung cancer; IC with SP142 antibody in breast, lung and urothelial tumors; and CPS with 22C3/SP263 assays in head and neck and urothelial carcinomas. On the other hand, for other malignancies, such as gastroenteric neoplasms, immunotherapy has been only recently introduced, often for particular histotypes, so specific guidelines are still lacking. Conclusions. PD-L1 immunohistochemical scoring is currently the basis for allowing many cancer patients to receive properly targeted therapies. While protocols supported by proven data are already available for many tumors, dedicated studies and clinical trials focusing on harmonization of the topic in other still only partially explored fields are surely yet advisable.

## 1. Introduction

Immune escape is one of the main ways adapted by tumor cells to evade the host’s immune response, thrive and then spread to distant organs. As well as in physiological inflammatory response, molecules of the CD28/CTLA4 family play a key role in this process [1,2]. Among these, programmed death protein 1 (PD-1) and its programmed death-ligand 1 (PD-L1) and 2 (PD-L2) are directly involved in cancer immune regulation and are considered one of the most relevant inhibitory checkpoints. Namely, the interaction between tumor-infiltrating lymphocytes (TIL) PD-1 and PD-L1 expressed on the surface of both tumor cells and other leukocytes triggers molecular pathways leading to reduced immune cell function [3]. For instance, PD-L1 downregulates cancer immune response by keeping foreign antigen-specific T cells from accumulating and reducing antigen-specific CD8+ T cell proliferation [4]. PD-1/PD-L1 blockage leads to the activation of T cells that can recognize and attack cancer cells [5,6]. Molecules able to inhibit the PD-1/PD-L1 axis may restore immunological reaction against tumor cells, and, on this basis, in the last decade, several monoclonal antibodies targeting both PD-1 (as Pembrolizumab and Nivolumab) and PD-L1 (including Atezolizumab, Avelumab and Durvalumab) were developed [7,8]. Such drugs have represented the milestones of modern cancer immunotherapy and are nowadays proficiently employed as the main therapeutic option for many kinds of tumors, often achieving far better results than conventional chemotherapy [9]. However, a wide percentage of patients do not respond to such immune checkpoint inhibitors (ICI) while potentially experiencing serious adverse drug reactions, so it is not easy to choose who immunotherapy should be administered to [10,11]. PD-L1 expression by tumor cells and TILs assessed by immunohistochemistry (IHC) is considered the most reliable predictive marker for response to immunotherapy in different tumor types [12,13,14,15,16,17]. Along with therapeutic monoclonal antibodies, several immunohistochemical assays have been created for PD-L1 evaluation. However, by using the commercially available clones against PD-L1, even striking divergent results may be obtained not only when considering tumors arising from different organs but also when performed on the same sample from the same patient, potentially greatly affecting eligibility for anti-PD-1/PD-L1 treatments [18]. In this view, it is mentioned that in clinical practice, neoplastic tissue can be obtained from surgical specimens, core biopsies and fine-needle aspiration often derived from different anatomical sites. Differences in expression between primary and metastatic samples, generally with higher PD-L1 expression in this latter [19], likely underlie tumoral clonal evolution and create further difficulty in choosing the most appropriate sample for PD-L1 testing. Similarly, several scoring systems evaluating various tumoral and immunological cellular compartments were developed for quantifying PD-L1 expression in different kinds of malignancies: among these, (i) the tumor proportion score (TPS) estimates the proportion of positive PD-L1 tumor cells among all viable tumor cells [20], (ii) the combined proportion score (CPS) is the ratio of the overall amount of positive tumoral and non-tumoral cells (lymphocytes and macrophages) to the total number of viable tumor cells multiplied per 100 [21], and (iii) the immune cell score (IC) refers to the area occupied by PD-L1 positive immune cells relative relatively to the entire tumor and peritumoral area [22].

These considerations notwithstanding, the aim of this work was to review and update the evidence on the application of the PD-L1 test in the situations most frequently requested by the pathologist, characterizing currently recommended indications, diagnostic platforms and scoring systems.

### 1.1. Lung Cancer

The current indications for PD-L1 expression assessment in lung cancer include both stage IV and locally advanced stage III neoplasms, with a feasible use of ICIs as a first or second line of treatment, either alone or in combination with chemotherapy or other ICIs [23]. Some ICIs can be used either in non-small cell lung cancer (NSCLC) such as adenocarcinoma and squamous cell lung carcinoma (Pembrolizumab) [24], while others even in small-cell lung cancer (SCLC) (Atezolizumab, Durvalumab) [25,26]. 

There are currently two scoring methods commonly used for PD-L1 in lung cancer:(1)TPS/TC (tumor cell score): percentage of tumor cells with membranous positivity, regardless of staining pattern and intensity, evaluated on at least 100 viable tumor cells.

PD-L1 expression is reported as TPS or TC depending on the ICI used, although they are considered identical in practical terms (Figure 1). The TC value is divided into cut-off groups: TC0: <1%; TC1: 1–4%; TC2: 5–49%; TC3: ≥50%;

(2)IC, considering a tumor area of at least 50 tumor cells, regardless of staining pattern and intensity. Like TC, the IC value is also divided into cut-off groups: IC0: <1%; IC1: 1–4%; IC2: 5–9%; IC3: ≥10%.

According to the registration trials, the Food and Drug Administration (FDA) approved the use of the aforementioned ICI as long as PD-L1 assessment is performed relying on specific clones and scoring systems, namely: (i) for Pembrolizumab, 22C3 clone (Dako) by TPS [27]; (ii) for Atezolizumab, SP142 (Ventana) by TPS or IC [28]; and (iii) for Durvalumab, SP263 (Ventana) by TPS [29]. Unlike FDA, EMA (European Medicines Agency) allows ICIs therapy regardless of the specific platform and antibody clone used for the immunohistochemical test specifically employed during the drug clinical trial.

### 1.2. Gastrointestinal Cancers

In the gastrointestinal/biliopancreatic system, the pattern of expression and the potential predictive role of immunotherapy-based regimens of PD-L1 was extensively studied. In this context, it had to be clarified that a real predictive role for this biomarker has been acknowledged for tumors of the esophagus, esophago–gastric junction and stomach [30]. Conversely, for intestinal cancers and for tumors of the pancreatobiliary system, PD-L1 has no recognized role in predicting the response to immunotherapy [30,31,32,33,34].

For tumors of the esophagus (both squamous cell carcinoma and adenocarcinoma), esophago–gastric junction and stomach, recent studies clarified the pivotal role of the 22C3 (Dako) and 28-8 (Dako) clones [35,36,37]. At the immunohistochemical level, the staining pattern should be assessed using the CPS. The CPS thresholds for considering a specific case as “positive” (i.e., eligible for immunotherapy) in this context represent a changing landscape, with current evidence suggesting a threshold of five for gastric tumors, also including those of the esophago–gastric junction, and of 10 for esophageal malignancies [35,36,37]. Given that this scenario is in constant and rapid evolution, the general suggestion for pathologists assessing PD-L1 in esophageal–gastric specimens is to provide the exact CPS score in the pathology report. The new multidisciplinary dimension of medicine may also suggest the possible discussion of the PD-L1 expression within a tumor board, including at least pathologists and oncologists. However, the most important indication is providing the oncologists with the exact CPS number, which is used by them for the potential choice of an immunotherapy-based regimen, taking into account the current indications of clinical practice at that specific time.

In the context of esophageal–gastric tumors, a recent study also investigated PD-L1 expression patterns in pre-neoplastic/dysplastic lesions, also indicating that biopsy material could be used in the real world as reliable for PD-L1 assessment. Of note, matching by grade of dysplasia, PD-L1-positive cases showed a higher prevalence in esophageal specimens compared with gastric ones. Moreover, PD-L1 expression was more prevalent and with higher positive scores in high-grade dysplasia and adenocarcinoma samples in comparison with low-grade dysplasia [38].

In intestinal/colorectal cancers, due to their intrinsic biology, the most important biovariables for assessing the eligibility of patients for immunotherapy are represented by microsatellite instability (MSI) and tumor mutational burden (TMB) [30]. In particular, the highest rate of response is reached when a given tumor simultaneously shows MSI and high TMB [39]. Although common, this situation is not always present and calls for an ongoing improvement in the selection of patients for immunotherapy. A recent study showed that a high PD-L1 expression in colorectal tumors might be associated with a poorer prognosis, but evidence has still not been demonstrated in terms of predictive values of such biomarker in this district [32].

In the pancreatobiliary system, the role of PD-L1 expression in assessing the eligibility of patients for immunotherapy is also limited. Differently from intestinal tumors, in such a district, MSI is rarely present, but currently, it represents the only acknowledged predictive criterion for immunotherapy [40,41,42]. In this context, high-TMB is also showing a promising profile along this line [43]. In the rare variant of pancreatic ductal adenocarcinoma called undifferentiated carcinoma with osteoclast-like giant cells, PD-L1 expression has been associated with a poor prognosis [44]. 

### 1.3. Breast Cancer

In breast cancer, PD-L1 expression is often associated with poor clinicopathologic features, high TIL count and triple-negative phenotype [45,46,47,48]. Thus far, PD-L1+ and treatment-naïve triple-negative breast cancer (TNBC) patients have been the most suitable candidates for ICI therapy in breast cancer [49]. Preliminary evidence regarding the clinical activity of pembrolizumab was firstly reported in the KEYNOTE-012 trial (NCT01848834). This study included previously treated, advanced TNBC patients showing an objective response rate (ORR) of 18.5% and a median time to response of 17.9 weeks (range, 7.3 to 32.4 weeks) [50]. Subsequently, the phase II KEYNOTE-086 (NCT02447003) examined the efficacy and safety of pembrolizumab monotherapy in two cohorts of patients. The first cohort included patients exposed to one or more prior lines of systemic treatment for metastatic disease regardless of PD-L1 expression, while the other cohort involved PD-L1+ patients with non-systemic anticancer therapy [51,52]. The PD-L1+ cohort showed an ORR of 23% (95% CI 14–36). This suggested a promising antitumor activity of pembrolizumab as first-line therapy for PD-L1+, metastatic TNBC [51,52,53].

Antibodies for PD-L1 assessment by IHC have different interpretation guidelines, and a variety of antibody clones and scoring systems are currently available (Figure 2). A post hoc analysis of IMpassion130 assessed the analytical concordance of PD-L1 evaluation by SP142, SP263 (Ventana) and PD-L1 22C3 (Dako) IHC assays [54,55,56]. PD-L1 positivity was considered as IC ≥ 1% for SP142 and SP263 assays and CPS ≥ 1 for the 22C3 assay. These findings showed that the 22C3 and SP263 assays identified more patients with PD-L1+ tumors (81% and 75%, respectively) compared to SP-142 (46%) [54]. However, while a lower number of cells were stained using SP142, the absolute benefit for the addition of atezolizumab for the samples that were only positive with 22C3 assay (progression-free survival (PFS): 1.7 months) or SP-263 (PFS: 1.6 months) was considered different in comparison with those that were only positive for SP-142 (PFS: 4.2 months) [57]. As the SP-142 antibody showed the greatest clinical benefit in the TNBC, PD-L1+ patient and Ventana PD-L1 (SP142), the IHC assay was the only approved companion diagnostic test for atezolizumab using the tumor-infiltrating IC score with a cut-off of 1% [58,59,60,61].

### 1.4. Urothelial Cancer

Several immunotherapeutic drugs targeting PD-1 and PD-L1 were approved by the FDA and EMA for the treatment of advanced and metastatic urothelial carcinoma (UC). The current guidelines for the treatment of advanced and metastatic UC indicate the possibility of prescribing ICI in the second line for patients that progressed during or after platinum-based therapy regardless of PD-L1 status [62]. On the other hand, for metastatic/advanced UC cisplatin-unfit patients, two ICI drugs, Keytruda (Pembrolizumab) and Tecentriq (Atezolizumab), were approved by FDA and EMA as first-line therapy, but their use is restricted to patients harboring a PD-L1 positive tumor [63,64]. 

This restriction was issued in 2018 based on preliminary results from Keynote-361 and IMvigor130 trials that showed a reduced PFS with Keytruda and Tecentriq compared with chemotherapy in patients with locally advanced or metastatic UC who had not received prior therapy and whose tumors had low expressions of PD-L1. In certain European countries, such as Italy, the only immunotherapy drug approved in first-line therapy is Pembrolizumab, but its use is not refunded by the National Health Service. Both drugs are only indicated in platinum unfit patients whose tumors are PD-L1 positive, as assessed by immunohistochemistry. Thus, PD-L1 IHC testing is now required in cisplatin-ineligible UC patients. According to the current recommendations and requirements of the FDA and EMA, PD-L1 status evaluation should be performed with the specifically approved PD-L1 assay (“companion diagnostics”) before administration of therapy with each of the PD-L1 inhibitors [64]. Namely, as for atezolizumab, PD-L1 positivity is defined as IC ≥ 5% measured by the Ventana PD-L1 SP142 assay. With regard to pembrolizumab instead, PD-L1 positivity is defined as a CPS of >10 using the Dako 22C3 platform (Figure 3). Tumor cells included in CPS scoring for UC are all the viable urothelial cancer cells of the invasive component as well as cells of the high-grade papillary carcinoma and of the carcinoma in situ component. Low-grade papillary carcinoma cells are excluded from both the numerator and the denominator.

The pre-analytical recommendation includes selecting a tissue block with at least 100 viable tumor cells of invasive UC, generating a fresh tissue section to prevent storage damage effects and using an on-slide positive control (tonsil tissue) [65]. The final report should include the following information: the assay performed and platform used, the readout parameter being reported (i.e., IC and/or CPS), and the clinically established cut-off category within which the result falls [66].

### 1.5. Kidney Cancer

PD-L1 immunoexpression in renal cell carcinoma (RCC) was analyzed by different clones and various cut-offs in different clinical and research settings [67]. Both TPS and CPS were studied: usually, a minority (7–15%) of clear cell RCCs harbor diffuse TPS scores (≥50%), while conversely, most of them (85%) show positive PD-L1 expression by using the CPS score (85%). The discrepancy is mainly due to the assessment of TIL and myeloid cells in the CPS, which increases the number of positive cells in the final score [68]. By focusing on the available clones, Ventana SP142 tends to label much more inflammatory than neoplastic cells, while Ventana SP263 mainly stains tumor cells, such as Dako 22C3. At a clinical level, the combination of Pembrolizumab with axitinib was approved in 2019 by the FDA for the first-line treatment of patients with metastatic renal cell carcinoma based on the results of the KEYNOTE-426 (NCT02853331) study [69]. This trial was conducted regardless of the PD-L1 expression status and revealed a 12-month overall survival rate of 90% for patients receiving pembrolizumab vs. 78% for those treated with sunitinib, along with improved PFS [70]. Furthermore, as pointed out by the KEYNOTE-564 (NCT03142334) study, adjuvant therapy with Pembrolizumab may be a chance for RCC patients with clear cell aspects characterized by intermediate/high relapse risk signatures, including sarcomatoid features or G4 ISUP/WHO nucleolar grading, as long as advanced pTNM staging (pT3 or pT4) [71]. However, albeit routinely employed in the US, immunotherapy adjuvant regimens have not been approved yet by national drug agencies in many European countries, including Italy, where adoption of ICI in this setting of patients may currently be only justified in controlled clinical trials.

### 1.6. Melanoma

The most important indication for the PD-L1 testing in melanoma patients comes from the results of the CheckMate067 study, a phase 3 trial comparing Nivolumab and Ipilimumab alone or combined in advanced melanoma patients [72]. In patients with an expression of PD-L1 ≥5%, the association of the two drugs showed a significantly better PFS. Although a cut-off of 5% was initially set, a follow-up study of the same cohort revealed a better stratification when a 1% lower positivity cut-off was used considered [73].

Different clones were variously employed in the literature. In the aforementioned CheckMate067 study, the 28-8 (Dako) was used. Conversely, in the Keynote006 trial combining different schedules of Pembrolizumab vs. the anti-CTLA4 Ipilimumab antibody, the 22C3 cloned (Dako) was the chosen testing antibody [74]. Nevertheless, a recent study compared the different available clones confirming both the reliability and the performances of all the 22C3, 28-8 and SP263 (Ventana) assays [75]. 

In the majority of previous and other studies investigating PD-L1 expression in melanoma [72,76], the TPS was used, but in 2016, a new “MEL score” was introduced, specifically created for melanoma, using the 22C3 clone (Dako) [77]. This score was based on the Allred Score derived from a scheme commonly used for hormone receptors in breast cancer and modified for melanoma. Namely, the percentage of positive cells was considered regarding both tumor cells and peritumoral mononuclear inflammatory cells intercalated within or closed to neoplastic nests. Inflammatory cells dispersed in the stroma distinct from tumor nests were instead excluded. In a six-tiered 0 to 5 scale, scores from 2 to 5, corresponding to ≥1% staining, were considered positive, whereas scores of 0 or 1, corresponding to <1% staining, were considered negative. Lately, the MEL score for assessment of PD-L1 staining was adopted by several trials with MEL score all using the 22C3 clone [78,79]. In conclusion, an assessment of PD-L1 in melanoma patients could be carried out using all the available clones. TPS with a 1% cut-off has been the most widely used score, but the MEL score, with a >2 cut-off, might also be used if the 22C3 clone is available.

### 1.7. Head and Neck Cancer

Recent trials investigating the efficacy of first-line immune checkpoint inhibition in recurrent and/or metastatic head and neck squamous cell carcinoma (HNSCC) showed that PD-L1 expression is associated with an increased ORR in patients with CPS ≥1 [80,81]. These studies allowed FDA in 2019 and EMA in 2020 to approve Pembrolizumab in recurrent and/or metastatic HNSCC in combination with platinum and fluorouracil regardless of PD-L1 status and in monotherapy in patients with CPS≥1 (Figure 4).

The registration trial was performed using the 22C3 clone (Dako) on Agilent autostainer link 48. Contrarily to FDA, EMA approved the immunotherapy in HNSCC regardless of the test (antibody and IHC platform) used, introducing the problem of harmonization between different types of antibodies and IHC platforms for evaluating the expression of PD-L1 with CPS in HNSCC [82,83]. Recent works highlighted a similar distribution for PD-L1 expression between 22C3 (Dako) and SP263 (Ventana) assays in whole sections and tissue microarray samples, although a greater number of false-positive results seem to be accustomed to this latter antibody [84]. Moreover, proper estimation of PD-L1 CPS in HNSCC may be hampered by intratumoral heterogeneity due to the specimens’ variability and effects of previously given treatments. In fact, it was shown that samples collected from different portions of a surgical resected HNSCC might harbor significantly differing CPS scores, ranging from 0 to 100 [85]. Therefore, such a finding is of major concern for bioptic material as a negative CPS scoring could potentially underlie a false negative result preventing patients from receiving appropriate immunotherapy. Additionally, not all patients diagnosed with HNSCC receive ICI as the first therapeutic option, so they might have received several combinations of classic chemotherapy and radiotherapy before starting PD-L1 inhibitors. It was reported that previously given therapies might cause PD-L1 expression both to increase [86] and to reduce [87], adding further difficulties in interpreting the results. Clearly, further work is needed to draw stronger conclusions about the interchangeability of PD-L1 assays in HNSCC. For a practical approach, it is advisable to re-biopsy and to re-evaluate PD-L1 CPS in cases reoccurring after chemo- and/or radiotherapy, even if the tumor had been PD-L1 negative at first diagnosis.

Placenta, tonsil and vermiform appendix tissues, as well as cancer cell lines, can be used as positive adequate quality controls. Diagnostic report for PD-L1 assessment in HNSCC should contain the patient’s clinical data, platform, immunohistochemical clone and controls used, type of score with clinical cut-off and the absolute value of PD-L1 as CPS.

### 1.8. Cytopathology

To date, all the available PD-L1 assays have been validated on formalin-fixed paraffin-embedded (FFPE) specimens; however, in the last decade, cytological samples have emerged as a valuable source of material for predictive biomarker testing. Therefore, an increasing amount of evidence has proven the feasibility of PD-L1 evaluation on cytological specimens by assessing specific pre-analytical issues, cyto-histological correlation and inter-observer reproducibility, in particular in the NSCLC setting.

As far as the pre-analytical phases are concerned, formalin-fixed cell block (CB) preparation is generally recommended, as FDA-approved or CE-marked PD-L1 IHC kits should be used for cytological samples processed according to the pre-analytical conditions provided by the kit [88]. Otherwise, appropriate protocols for cytological materials should be validated separately for any type of preparation (direct smears or liquid-based cytology) [89]. 

As for the analytical phase, the literature data showed good results in terms of adequacy rate of cytological samples for PD-L1 evaluation [90]; moreover, the role of rapid on-site evaluation (ROSE) in the improvement of CB adequacy and cellularity should not be overemphasized. 

The feasibility of PD-L1 evaluation on cytological material is also confirmed by the concordance values between matched histological and cytological samples: in fact, an overall concordance rate of 88.3% at a clinically relevant cut-off of TPS > 1% and of 89.7% for specimens with TPS ≥ 50% were reported in NSCLC setting [91]. Nonetheless, the assessment of PD-L1 on cytological samples presents several challenges, such as unspecific staining in the background or in tridimensional groups; consequently, data on interobserver agreement showed a more evident variability in cytological preparations than in histological samples [92]. 

As opposed to TPS assessment, the evaluation of PD-L1 expression using CPS has not been well established on cytological material, mainly because of challenges in identifying tumor-associated immune cells in the absence of tissue architecture. Although still limited, the literature data showed that the evaluation of CPS on cytological samples is feasible, especially in the case of positive PD-L1 expression [93,94]. 

In conclusion, cytological material is a reliable source for PD-L1 evaluation; however, pre-analytical protocol validation and focused expertise in IHC interpretation on cytological preparation are recommended.

### 1.9. Hematological Disorders

Although lymphocytes, macrophages and other immune cells have historically represented the basis that led to the development of modern ICI, anti-PD-L1 therapy has been less extensively studied in hematological neoplasms, especially lymphomas, than other solid malignancies [95]. In this setting, actually, the widest proportion of data comes from those lymphomas particularly enriched with a background of lymphocytes, histiocytes, plasma cells and granulocytes, which act like a tumor-related inflammatory infiltrate. Among these, classic Hodgkin lymphoma (CHL) is indeed the most investigated disease, as PD-L1 is overall expressed in 70–80% of CHL cases, labeling both Reed–Sternberg cells and TIL [96]. Genetic alterations of the chromosome locus 9p24.1 mapping the *PD-L1*, *PD-L2* and *JAK2* genes, such as polysomy, copy gains and amplification, are the main mechanism underlying upregulated PD-L1 expression in CHL [97]. JAK/STAT signaling pathway activation by the Epstein–Barr virus’ (EBV) LMP-1 protein was claimed as the PD-L1 trigger in 9p24.1 diploid CHL cases [98]. Nivolumab and pembrolizumab were proven to be effective in patients with relapsed/refractory CHL, achieving an ORR of 60-70% in several clinical trials [99]. Thus, in 2016 and 2017, respectively, the FDA approved nivolumab and pembrolizumab for the treatment of patients with relapsed/refractory CHL.

Lower PD-L1 levels were instead reported in diffuse large B cell lymphoma (DLBCL), the most common type of non-Hodgkin lymphoma in adults, with only 10–24% of DLBCL cases being positive for PD-L1 [100], likely due to less frequent alterations of chromosome 9p24.1 than CHL. Higher rates of PD-L1 expression were reported in EBV+ DLBCLs [100], but it is not generally recommended to treat unselected DLBCL patients with PD-1/PD-L1 inhibitors. To date, PD-1/PD-L1 blockage therapy has been approved only for primary mediastinal large B cell lymphomas, a rare variant of DLBCL carrying 9p24.1 copy number gains in 29-55% of the cases [101], where pembrolizumab administration have shown encouraging results in clinical trials [102].

Finally, the PD-L1 pathway has also been studied in anaplastic large-cell lymphoma (ALCL), a T-cell disorder harboring *ALK* gene translocations in more than 80% of the cases [103]. Several studies reported strong immunohistochemical PD-L1 expression in ALCL [104,105], likely linked to both ALK-related and ALK-unrelated upregulation of the *STAT3* gene. Similarly, a dramatic and durable response to PD-1 blockade in patients was observed in some case reports [106,107]. However, as PD-1 signaling inhibition may accelerate the growth of T-cell lymphomas due to physiological PD-1 block of T cell proliferation [108], there is a concern regarding broad anti-PD-1/PD-L1 therapy in T-cell lymphomas, and treatment with such drugs is only recommended in highly selected cases.

In summary, while the immune biology of lymphoid neoplasms has helped to identify specific lymphoma types potentially vulnerable to PD-1/PD-L1 inhibitors, the future of inhibitors of this pathway in hematological disorders is still unclear. Widening the knowledge of underlying molecular mechanisms will also hopefully lead to broader adoption of such drugs in this setting of human malignancies, eventually integrating with highly effective therapies such as CAR-T cells.

Specific indications, available clones and routinely used scoring systems approved for PD-L1 assessment among different organs are summarized in Table 1.

## 2. Conclusions and Perspectives

Immune checkpoint inhibitors have represented a disruptive breakthrough in cancer therapy and, among these, drugs targeting the PD-1/PD-L1 indeed own a key role in modern immunotherapy. However, despite encouraging clinical results, not all patients can benefit from immunotherapy. To date, immunohistochemical assessment of PD-L1 expression is still considered the most reliable marker predictive of response to available immunotherapeutic drugs. Several platforms, PD-L1 clones and scoring systems were then developed to address such needs. However, while consolidated data are available for different types of tumors, such as lung cancer, which have historically been the milestones of immunotherapy application, evidence is still evolving for other fields of pathology where ICI represent an emerging therapeutic option. Therefore, further dedicated studies are required to address the harmonization of employed clones, scores and interobserver variability, potentially exploiting fast-sharing tools such as digital pathology [109,110,111], in order to ultimately allow patients to obtain the most proper effective and safe clinical management possible. Indeed, the disruptive progression of this latter technology has recently revolutionized many fields of surgical pathology, and computer-based automatic learning algorithms are currently widely available. As for immunohistochemistry, while firstly employed for the evaluation of breast cancer biomarkers [112], artificial intelligence has also been exploited for PD-L1 assessment. Despite not satisfactory initial performance linked to difficulty in classification and count of tumor-infiltrating immune cells [113], recently published papers witnessed encouraging results compared to conventional scoring by differently experienced physicians [114]. It is then foreseeable that in the very next future, further development of these algorithms and artificial intelligence would be proficiently incorporated into daily diagnostic workflow, contributing to assisting pathologists in routinely PD-L1 scoring.

## Figures and Tables

**Figure 1 jpm-12-01073-f001:**
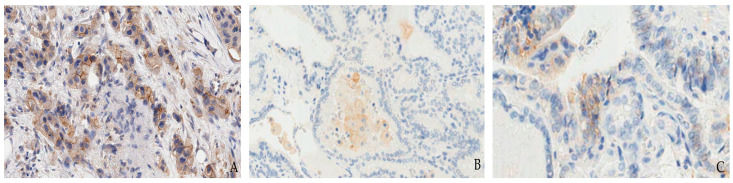
Tumor proportion score (TPS) in lung adenocarcinoma. TPS 80% (**A**), TPS 0% with positive alveolar macrophages representing internal control (**B**), and TPS 30% (**C**).

**Figure 2 jpm-12-01073-f002:**
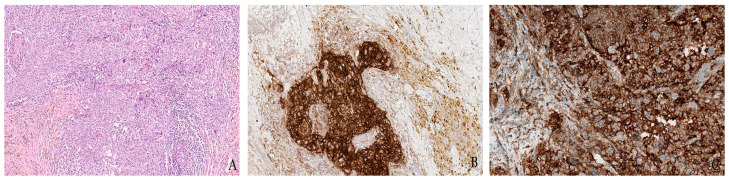
Representative microphotographs of a poorly differentiated (G3) triple-negative breast cancer of no special type (**A**) showing the different patterns of PD-L1 immunohistochemical staining using a 22C3 clone on a Dako Autostainer Link 48 (**B**) and an SP263 clone on a Ventana BenchMark Ultra (**C**) for combined proportion score (CPS) testing.

**Figure 3 jpm-12-01073-f003:**
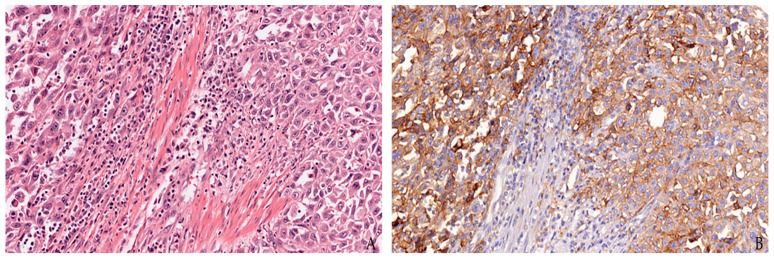
Representative image for PD-L1 immunostaining in urothelial carcinoma: hematoxylin and eosin (**A**) and PD-L1 22C3 clone by Dako (**B**).

**Figure 4 jpm-12-01073-f004:**
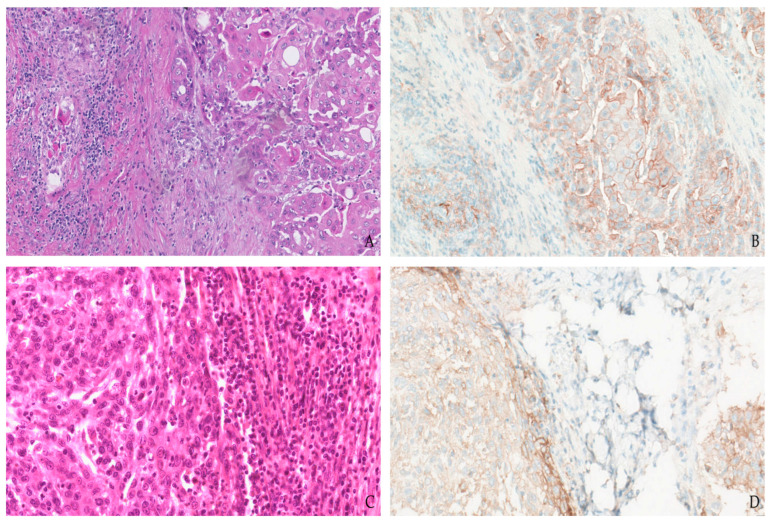
Combined proportion score (CPS) in head and neck cancer. Hematoxylin and eosin stain (**A**,**C**) show two examples of squamous cell carcinoma rimmed by tumor-infiltrating lymphocytes (TIL). Immunohistochemical analysis of PD-L1 expression with 22C3 clone labeling almost all tumor cells and a noteworthy proportion of TIL, resulting in a CPS = 80 (**B**). Assessment of PD-L1 staining with the SP263 antibody revealing homogenous membranous expression by neoplastic cells and partial positivity of peripherical TIL for an overall CPS = 75 (**D**).

**Table 1 jpm-12-01073-t001:** Currently approved therapeutic indications, clones and scoring systems for immunohistochemical evaluation of PD-L1 status.

Tumor	Indications	Scoring System (Clones) and Therapeutic Guidelines
Lung cancer	1L/2L in stage IV NSCL or diffuse SCLC	TPS ≥ 1% (22C3, SP142, SP263) and IC ≥ 10% (SP142) *
GE cancer	1L or following lines in stage IV	CPS ≥ 1 (22C3, 28-8)
Colon and pancreas cancer	1L or following lines in stage IV MSI-H	IC ≥ 1% (28-8) (registration trial Check-Mate 142)
Breast cancer	1L or following lines in stage IV TNBC	IC ≥ 1% (SP142)
Urothelial carcinoma	1L platinum-unfit, 2L platinum-fit both in stage IV	CPS > 10 (22C3) and IC ≥ 5% (SP142)
Kidney cancer	1L in stage IV RCC	Therapy given regardless of PD-L1 status (22C3, SP142, SP263)
Melanoma	1L in stage IV melanoma	TPS ≥ 1% (22C3, 28-8, SP263) and MEL score > 2 (22C3)
HNSCC	1L in recurrent or stage IV HNSCC +/− platinum	CPS ≥ 1 (22C3, SP263) or regardless of PD-L1 status (+ platinum)

* use of specific scoring systems for each clone is recommended by FDA but not by EMA. Abbreviations: 1L: first line, 2L: second line, NSCLC: non-small cell lung cancer, SCLC: small cell lung cancer, TPS: tumor proportion score, IC: immune cell score, GE: gastro-esophageal, CPS: combined proportion score, MSI-H: high microsatellite instability, NA: not available, TNBC: triple-negative breast cancer, RCC: renal cell carcinoma, HNSCC: head and neck squamous cell carcinoma.

## Data Availability

Not applicable.

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
