# Peer review of "Atlas of PD-L1 for Pathologists: Indications, Scores, Diagnostic Platforms and Reporting Systems"

_jpm, 2022, doi:10.3390/jpm12071073_

Round 1

Reviewer 1 Report

This review article briefly summarized the current status of PD-L1 as a biomarker for immune checkpoint inhibitor treatment. The review provides useful information on various scoring systems and diagnostic platforms for PD-L1 expression that will be a common interest for oncologists and pathologists. The authors have focused on several solid cancers but including discussions on hematological malignancies would improve readership.    

Reviewer 2 Report

This manuscript by Marletta et al. summarizes the evidence on the application of the PD-L1 test in the situations most frequently requested by the pathologist, characterizing currently recommended indications, diagnostic platforms and scoring systems. I recommend a minor revision.

1. The "INTRODUCTION" section should be enriched.

2. I suggest authors to add a representative figure in each subsection in the manuscript.

3. Authors should provide their own perspectives in "CONCLUSIONS AND PERSPECTIVES" section.
